# Gate-controlled conductance switching in DNA

Limin Xiang[1,2], Julio L. Palma[1,2,3], Yueqi Li[1,2], Vladimiro Mujica[2], Mark A. Ratner[4] & Nongjian Tao[1,5]

Extensive evidence has shown that long-range charge transport can occur along double helical DNA, but active control (switching) of single-DNA conductance with an external field has not yet been demonstrated. Here we demonstrate conductance switching in DNA by replacing a DNA base with a redox group. By applying an electrochemical (EC) gate voltage to the molecule, we switch the redox group between the oxidized and reduced states, leading to reversible switching of the DNA conductance between two discrete levels. We further show that monitoring the individual conductance switching allows the study of redox reaction kinetics and thermodynamics at single molecular level using DNA as a probe. Our theoretical calculations suggest that the switch is due to the change in the energy level alignment of the redox states relative to the Fermi level of the electrodes.

[1] Biodesign Center for Biosensors and Bioelectronics, Biodesign Institute, Arizona State University, Tempe, Arizona 85287, USA. [2] School of Molecular Sciences, Arizona State University, Tempe, Arizona 85287, USA. [3] Department of Chemistry, The Pennsylvania State University, Fayette, The Eberly Campus, 2201 University Drive, Lemont Furnace, Pennsylvania 15456, USA. [4] Department of Chemistry, Northwestern University, Evanston, Illinois 60208, USA. [5] School of Electrical, Computer and Energy Engineering, Arizona State University, Tempe, Arizona 85287, USA. Correspondence and requests for materials should be addressed to N.T. (email: njtao@asu.edu).

DNA is a unique molecule not only because of its role in living systems but also due to its double helical structure with π-electron stacking of the base pairs that has inspired many to explore DNA as a molecular wire[1–12]. In addition, recent advances have made it possible to design and synthesize DNA with programmable three-dimensional nanostructures[13], which have further stimulated efforts to study DNA as device building blocks. Extensive theoretical and experimental works have indeed established that long-range charge transport can occur along double helical DNA via the overlapping π molecular orbitals of the stacked bases[1,3,14,15]. While short-range charge transport in DNA has been attributed to non-resonant tunnelling[4,7], long-range charge transport is due to hopping between DNA bases[3,14,15], dominated by Guanine because its highest occupied molecular orbital (HOMO) level is closest to the electrode Fermi level among the four DNA bases[16]. However, to further understand charge transport in DNA and to explore possible device applications, one wishes to electrically switch DNA conductance between different states with an external field. This has not yet been demonstrated. Here we report a molecular switch by replacing a DNA base in double helical DNA with a redox group. By controlling the electrochemical (EC) gate voltage, we reversibly switch the DNA conductance between two discrete levels via the redox group. The population of the reduction and oxidation states follows the Nernst equation, and analysis of the conductance switching allows determination of the rate constants of the redox process at the single-molecule level. Theoretical calculation shows that the conductance switching arises from a change in the molecular energy alignment associated with the redox state switching.

## Results

**Structures and conductance measurements of DNA.** To switch DNA conductance, we replaced one of the regular DNA bases with anthraquinone (Aq), a redox group that can be reversibly oxidized and reduced (Fig. 1b). Nuclear magnetic resonance and molecular dynamics analysis of a similar structure suggest that the Aq moiety stacks on the adjacent GC base pair, and the non-paired Guanine ring rests atop the Aq ring[17]. This conformation is highly stable as indicated by the melting temperature increasing effect after this modification[17,18]. It also allows overlapping of the anthraquinone molecular orbitals with those of the neighbouring bases, thus providing a continuous π–π stacking pathway along DNA for efficient charge transport (Fig. 1b). To ensure high stability of double-stranded DNA and high charge transport efficiency, we choose the DNA sequences that consist of GC base pairs only. As a control experiment we also studied DNA without the Aq moiety. We refer the redox modified DNA as Aq-DNA, and unmodified DNA as u-DNA (Fig. 1b). Gel electrophoresis confirmed that both Aq-DNA and u-DNA form double helical structure, and no other structures were present under the experimental conditions (Supplementary Fig. 1).

We measured the DNA conductance using a scanning tunnelling microscope (STM) break junction technique[19,20]. The technique used a gold tip coated with wax to minimize ionic conduction[7], and a gold substrate (Fig. 1a). The tip was repeatedly brought into and pulled out of contact with the substrate in a buffer solution containing the DNA molecules, during which the current between the tip and substrate was continuously monitored. Individual current versus tip-substrate distance traces were recorded during the pulling process (Fig. 1c) and a plateau in the current traces signalled the formation of a single gold–DNA–gold molecular junction. Thousands of

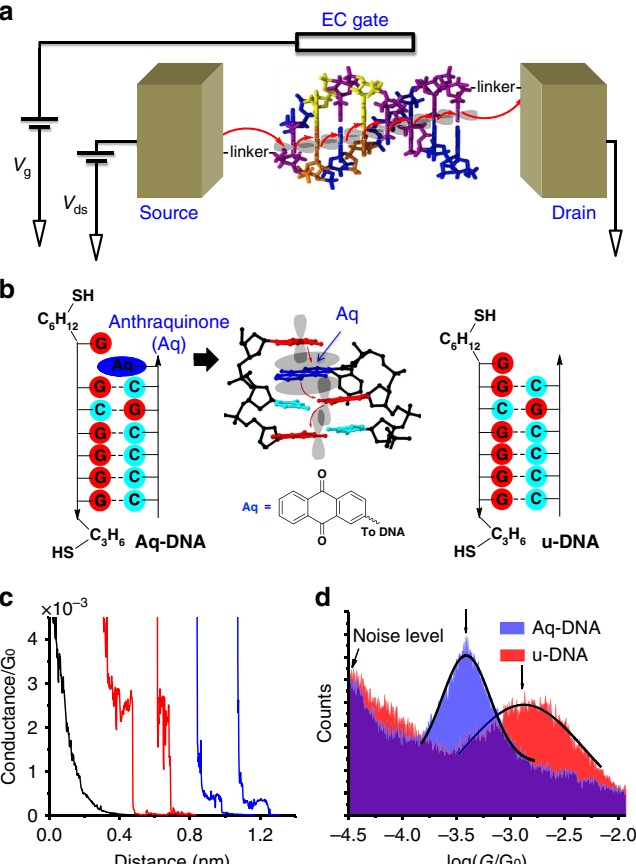

**Figure 1 | EC gate control of DNA conductance.** (**a**) Illustration of the experiment, where the source and drain electrodes are the STM tip and substrate, and EC gate is a silver electrode inserted in the solution. A DNA molecule bridged between the source and drain electrodes via the thiolate linker groups, where charge hops from one base to the next (red arrows) via overlapping π-orbitals. The source-drain bias ($V_{ds}$), and the EC gate voltage ($V_g$) are controlled independently. (**b**) From left to right: redox modified DNA (Aq-DNA), where a base was replaced with an anthraquinone (Aq) moiety (highlighted in blue) at the 3′-end of a DNA strand (see chemical structure in Supplementary Fig. 1a); three-dimensional structure (PDB ID: 2KK5, results are from nuclear magnetic resonance study[17]) shows that the Aq moiety intercalated in between the two Guanine bases on the other strand acts as a hopping site (red arrows) with its π-orbital overlapping with those from adjacent bases. Aq moiety is shown in blue. Picture is created from 2KK5 in PDB with JSmol software. DNA without the Aq moiety (u-DNA) was studied as control. Both Aq-DNA and u-DNA contain a strand terminated with thiolated linkers at the 3′- and 5′-ends for contact with the source and drain electrodes. (**c**) Representative current–distance traces (current converted to conductance) of Aq-DNA (blue) and u-DNA (red) in aqueous solution, showing plateaus originated from the formation of the DNA junctions. Control experiments performed in the absence of DNA molecules showing smooth exponential decay (black trace). (**d**) Conductance histograms of Aq-DNA (in blue) and u-DNA (in red), showing the difference in the conductance peaks. The peak was fitted with a Gaussian distribution and the peak position was taken as the conductance.

the current traces (∼4,000) were collected, and used to construct a conductance histogram (Fig. 1d), where the peak position indicates the most probable conductance value of the DNA molecules.

We first measured the conductance of both Aq-DNA and u-DNA without applying EC gate voltage, and found that

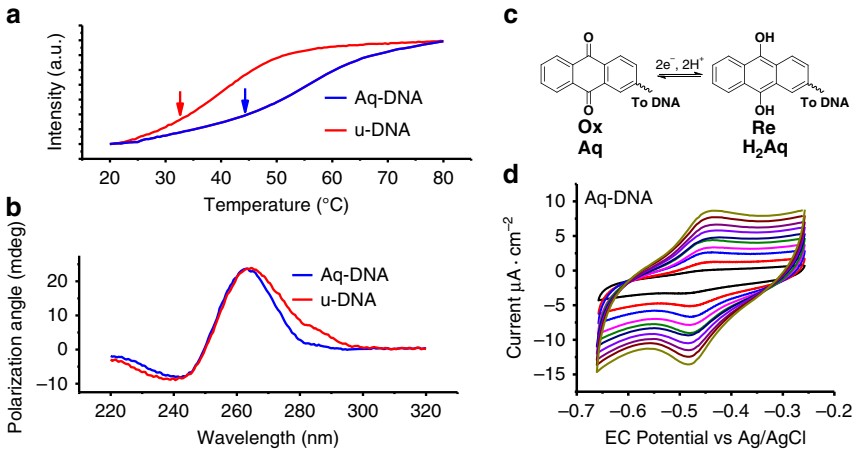

**Figure 2 | Characterizations of DNA molecules.** (**a**) Melting temperature curves for Aq-DNA and u-DNA with ultraviolet–vis absorption spectroscopy, where arrows indicate the melting points. (**b**) Circular dichroism study for Aq-DNA and u-DNA, where the negative band at around 245 nm, and positive band at ∼265 nm indicates double helical structure for both Aq-DNA and u-DNA. (**c**) Reversible redox reaction of the anthraquinone moiety involving two electrons in aqueous solution. (**d**) Cyclic voltammograms of Aq-DNA immobilized on gold substrate with potential sweeping rate varying from 0.01 (black line) to 0.1 V·s$^{-1}$ (green line) showing oxidation and reduction peaks.

the conductance values for Aq-DNA and u-DNA were $4.0 \pm 0.2 \times 10^{-4}$ G$_0$ and $14 \pm 1 \times 10^{-4}$ G$_0$, respectively, where G$_0 = 7.748 \times 10^{-5}$ S, which is the conductance quantum (Fig. 1d). These values are within the conductance range of DNA reported by others using similar thiolate linker groups[21–23], but larger than the conductance of anthraquinone-based molecular wire[24–28] and other conjugated systems[29], such as 4, 4′-bipyridine[30]. The transport mechanism in the present anthraquinone-DNA system is thermal activated hopping via anthraquinone and stacked bases, which decays with the molecular length slowly[31,32] compared with anthraquinone and other tunnelling systems[24–28]. Control experiments in the buffer solution without DNA molecules, and with single-stranded DNA terminated with two thiolate linkers (Supplementary Fig. 2) shown in Fig. 1b did not reveal peaks in the conductance histogram.

The observed conductance difference between Aq-DNA and u-DNA indicates that the Aq moiety was intercalated into the base pairs in DNA, as shown in literature[17] (Fig. 1b). To further verify this conclusion, we performed melting temperature analysis on both Aq-DNA and u-DNA (Fig. 2a). The melting temperature of Aq-DNA (44 °C) is much higher than that of u-DNA (32 °C), indicating that the intercalation of Aq stabilizes the ending base pairs[17,18]. Despite the stabilization effect, Aq-DNA retains the double helical structure close to B-form structure as shown by the circular dichorism spectra of Aq-DNA and u-DNA (Fig. 2b).

**Two conductance states controlled by gate voltage**. To switch the DNA conductance, we controlled the redox state of Aq-DNA by inserting a silver electrode into the solution (Fig. 1a)[33,34]. The silver electrode serves as an EC gate, and the tip and substrate serve as source and drain electrodes. The gate voltage and the source-drain bias were controlled using the standard four-electrode EC configuration with a platinum coil as an auxiliary electrode (not shown in Fig. 1a) in addition to the tip, substrate and silver quasi-reference electrodes. We first characterized redox-switching properties of Aq-DNA immobilized on the gold substrate with cyclic voltammetry, a widely used EC technique that measures EC current while linearly sweeping the substrate potential back and forth. The

measured cyclic voltammograms show a peak in the forward potential sweep, and a negative peak in the reverse potential sweep (Fig. 2d), corresponding to reversible oxidation and reduction of the Aq group in Aq-DNA (Fig. 2c). The redox potential of Aq-DNA, determined by taking the average of the oxidation and reduction peak potentials, is − 0.46 V versus Ag/AgCl reference. This finding is consistent with that reported for other anthraquinone-DNA intercalation complexes[35,36]. The separation between the oxidation and reduction peaks increases with potential sweeping rate and the peak heights increase linearly with the sweep rate (Supplementary Fig. 3b,c), indicating that the Aq-DNA was immobilized on the gold substrate. From the areas of the oxidation and reduction peaks, we determined the Aq-DNA surface coverage to be $1.48 \pm 0.03$ pmol·cm$^{-2}$, which is within the range of thiolate-modified DNA molecules on gold electrodes measured by EC methods[37]. This corresponds to 112 nm$^2$ per DNA molecule, which is relative low compared with the surface coverage of a compact layer consisting of DNA aligned vertically with a small tilt angle.

After characterizing the conductance and redox property of Aq-DNA, we studied conductance switching of Aq-DNA by controlling the gate voltage ($V_g$). For simplicity, we quoted the gate voltage with respect to the redox potential of Aq-DNA, at which the chances of oxidation and reduction are equal. At each gate voltage, we performed repeated STM break junction measurement in a nitrogen atmosphere, and constructed a conductance histogram from thousands of (∼4,000) individual conductance traces. When the gate voltage is well above 0 V, most Aq-DNA molecules are in the oxidation state (Fig. 3d) and the conductance histogram shows a peak at $3.7 \pm 0.5 \times 10^{-4}$ G$_0$ (Fig. 3a), which is the conductance of Aq-DNA in the oxidation state. Lowering the gate voltage towards 0 V, a second peak at a higher conductance value ($30 \pm 3 \times 10^{-4}$ G$_0$) appears (Supplementary Figs 4 and 5 and Supplementary Table 1). We attribute the high conductance peak to Aq-DNA in the reduction state. The high conductance peak increases in height, while the low conductance peak decreases in height with decreasing the gate voltage, which is expected as increasing number of Aq-DNA became reduced (Fig. 3e). The high and low conductance peaks reach a similar height when the gate voltage is 0.0 V, corresponding to an equal number of Aq-DNA in

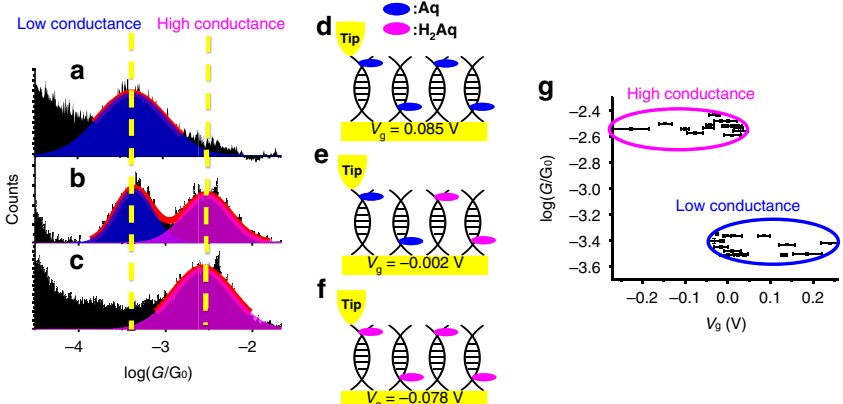

**Figure 3 | Two conductance states of Aq-DNA under gate voltages.** (**a**–**c**) Conductance histograms of Aq-DNA with the gate voltage set above (0.085 V in **a**), at ( − 0.002 V in **b**) and below ( − 0.078 V in **c**) the redox potential, where the red curves are Gaussian fits to the conductance peaks. (**d**–**f**) Populations of Aq-DNA in the oxidation (Aq) and reduction (H$_2$Aq) states at the corresponding gate voltages. (**g**) Conductance values at different gate voltages showing two discrete conductance states, high (circled with magenta line) and low (circled with blue line) conductance states. Conductance error is determined by the fitting error of the Gaussian distribution. Gate voltage error is determined by the variations in the quasi-reference electrode (Supplementary Fig. 3a).

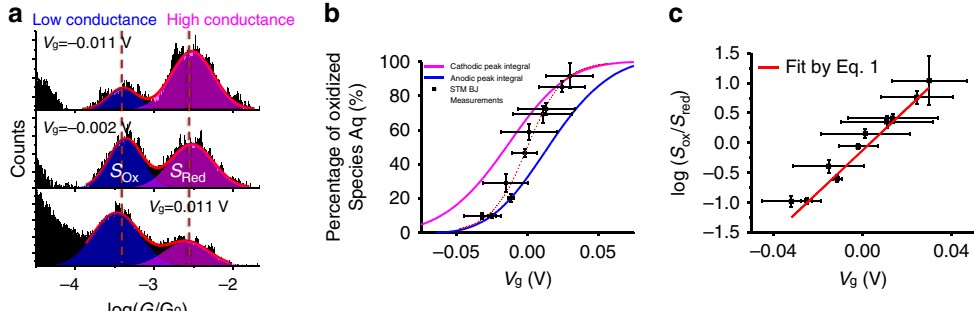

**Figure 4 | Thermodynamic analysis of the two-level conductance switching.** (**a**) Conductance histograms at different gate voltages ($V_g$), where the red lines are Two-Gaussian fits, from which the areas of the low ($S_{ox}$) and high ($S_{red}$) conductance peaks are determined. (**b**) Percentage of Aq-DNA in the oxidized form versus gate voltage obtained from the reduction (magenta curve) and oxidation (blue curve) peaks in the cyclic voltammograms with a sweeping rate of 0.1 V·s$^{-1}$, and the high and low conductance peaks in the conductance histograms (black points). Red dashed line is guide to eye. (**c**) Log($S_{ox}/S_{red}$) versus gate voltages ($V_g$) and fitting of the data with the Nernst equation (red line). Peak area error is determined by the fitting error of the Gaussian distribution. Gate voltage error is determined by the variations in the quasi-reference electrode (Supplementary Fig. 3a).

the reduction and oxidation states (Fig. 3b). Further decreasing the gate voltage below 0 V, the high conductance continues to increase while the low conductance peak decreases and eventually disappears (Fig. 3c) as all the molecules become reduced (Fig. 3f). We used the fitting errors (Gaussian fitting) in each of the histograms at different gate voltages as the experimental errors[38]. The broad distribution (the width in the Gaussian fit) is an inherent property of single-molecule measurements originated from the variation in the molecule–electrode contact coupling, and dependence of the conductance on the couplings[39,40], rather than an experimental error. This broad distribution is commonly observed for single molecular measurements of DNA[15,21,41].

The experiment described above shows that one could switch the DNA conductance between two levels by controlling the Aq redox state. Aq-DNA in the reduction state is nearly an order of magnitude more conductive than that in the oxidation state. Another interesting observation shown in Fig. 3a–c and Supplementary Figs 4 and 5 is that despite the sensitive dependence of the peak heights in the conductance histograms with the gate voltage, the peak positions change little with the gate voltage. This observation indicates that Aq-DNA takes two discrete conductance values that correspond to the oxidation and reduction states, and the gate voltage can only switch the conductance between the two values as Aq-DNA is either oxidized or reduced. The switching of conductance between two discrete levels is more clearly shown in the plot of conductance versus gate voltages in Fig. 3g and Supplementary Fig. 5d. Previous studies of redox molecules, including anthraquinone, typically show continuous changes of conductance with the gate voltage[20,24,25,33,42–45], and redox switching of the conductance between two discrete levels has not been reported before. We will return later to the mechanism and implication of this discrete conductance switching in Aq-DNA.

To confirm that the gate switching is due to the Aq moiety rather than DNA, we carried out a control experiment with u-DNA, and did not observe any significant changes in the conductance histogram over the same gate voltage range (see Supplementary Fig. 6). A previous study of gate dependent measurement of DNA conductance also failed to detect conductance switching in regular double helical DNA[46].

The control experiment shows that the two-level conductance switching was originated from the redox species Aq moiety. Aq-DNA and u-DNA have different conductance values, but a similar stretching length (0.12 nm, Supplementary Fig. 7). Stretching length is the average distance over which one can stretch a molecular junction before it becomes mechanically unstable, which is measured from the plateau length in the conductance trace as marked in Supplementary Fig. 7. The short stretching length observed for both Aq-DNA and u-DNA is consistent with reports in literature, which is attributed to force-induced melting of DNA[47,48].

**Thermodynamic study of the redox states.** We further analysed the gate voltage dependence of the peaks in the conductance histograms of Aq-DNA by calculating the peak areas at different gate voltage (Fig. 4a). This analysis allows us to determine the relative probability of Aq-DNA in the oxidation and reduction states at each gate voltage because the peak area is expected to be proportional to the number of Aq-DNA in the corresponding states[49] (Supplementary Note 1). The probability of oxidized Aq-DNA versus the gate voltage shows a sigmoidal dependence (black points in Fig. 4b). The result shown in Fig. 4b was obtained from the statistical analysis of single-molecule measurement, which can be compared with the cyclic voltammetry that measures a large number of DNA molecules. As shown in Fig. 2d, the cyclic voltammetry reveals oxidation and reduction peaks, and integration of the peak areas provide the amount of charge transfer ($Q$). Knowing that the number of charge involved in each oxidation (reduction) event for Aq moiety is 2, we determined the percentage of the molecules in the oxidation state at different gate voltages[50], and the result is plotted together with the single-molecule data from the conductance histograms in Fig. 4b (see Supplementary Fig. 3d–f for details). Note that forward (blue curve) and reverse (magenta curve) gate sweeping curves display hysteresis, but both have the sigmoidal shape. The hysteresis (separation between oxidation and reduction peaks) decreases with the potential sweeping rate (Supplementary Fig. 3b), indicating that the hysteresis is at least partially because the charge transfer between the electrode and Aq moiety is relatively slow compared with the sweeping rate[43]. The data obtained from the single-molecule measurement falls in between the forward and reverse gate sweeping curves. This is expected because each data point was measured by holding the gate at a fixed voltage, corresponding to an extremely slow scanning rate.

The relative probability of Aq-DNA in the oxidation and reduction states at thermal equilibrium is expected to follow the Nernst equation,

$$E = E_{\text{ox/red}} + 2.303 \frac{RT}{nF} \log\left(\frac{\Gamma_{\text{ox}}}{\Gamma_{\text{red}}}\right) \quad (1)$$

where $E$ is the applied potential and $E_{\text{ox/red}}$ is the redox potential of anthraquinone (gate voltage: $V_g = E - E_{\text{ox/red}}$), $n$ is the number of charge units transferred per molecule, $R$ is the universal gas constant, $T$ is the temperature in K and $F$ is the Faraday constant. $\Gamma_{\text{ox}}$ and $\Gamma_{\text{red}}$ are the surface concentration of the DNA molecules in the oxidation and reduction states, which are proportional to the equilibrium probabilities of the oxidized and reduced species, and can be determined from the areas of oxidation and reduction peaks ($S_{\text{ox}}$ and $S_{\text{red}}$) in the conductance histograms. Figure 4c plots $\log(S_{\text{ox}}/S_{\text{red}})$ versus gate voltage ($V_g$), and fitting the data with Nernst equation yields $n = 2.0 \pm 0.4$. This value is expected for anthraquinone, which is a prototypical reversible redox species with a two-electron transfer event. The agreement between the single-molecule conductance

measurement and the Nernst equation provides further evidence that the observed conductance switching in Aq-DNA is controlled by the redox state of the Aq moiety.

**Kinetic study of the redox states.** The conductance histogram analysis described above provides averaged properties of single DNA conductance switching events. To further understand conductance switching in DNA, we also studied the individual conductance switching events by monitoring the DNA conductance ($G$) over time ($t$) at different gate voltages. We first detected a plateau regime in a conductance-distance trace (wine arrows, Fig. 5a) at a fixed gate voltage[51], which corresponds to the formation of a DNA junction between the tip and substrate electrodes, and then froze the tip in position while recording the conductance ($G$) versus time ($t$) for 0.1 s.

Figure 5b shows four representative $G$–$t$ curves at a gate voltage of $0.000 \pm 0.005$ V, each starts when the conductance is at the high conductance level, or Aq-DNA in the reduction state. The four $G$–$t$ curves illustrate three types of conductance changes over time. Type 1 (30–40% probability): the conductance fluctuates but these changes are not large enough to be attributed to the switching of the redox state within the 0.1 s time window (black curve). This type of small conductance fluctuations is commonly observed in single-molecule junctions, which are attributed to the atomic scale rearrangement in the molecule–electrode contact[39,40]. Type 2 (20%): the conductance drops abruptly to the noise level, signalling the breakdown of a molecular junction (red curve). This type is also a characteristic of single-molecule junctions, arising from the finite lifetime in the molecule–electrode contact[52,53]. Type 3 (40–50%): the conductance switches from the high level to a lower level (blue curve), or switches back and forth between the high and low levels (magenta curve). The high level conductance is about an order of magnitude higher than the low level conductance. This two-level conductance switching is a unique feature of Aq-DNA, which measures reversible switching of the molecule between oxidation and reduction states. To confirm this conclusion, we performed control experiment with u-DNA and did not observe Type 3 switching behaviours (Supplementary Fig. 8).

To further analyse the two-level transient conductance switching, we constructed a two-dimensional (2D) $G$–$t$ histogram for Types 1 and 3 conductance curves (Fig. 5c). The 2D histogram reveals two distinct bands at the high and low conductance levels, which confirms the two-level switching of the Aq-DNA redox state discussed earlier. When the gate voltage is close to zero, one expects equal probabilities of finding Aq-DNA in the high and low conductance states, as expected by the Nernst equation (see Supplementary Note 2 for further discussions). Figure 5d shows the conductance histogram at $t = 0.0$ and 0.1 s, which indeed shows that if the molecule starts at the high conductance level (reduction state), the probabilities of high (reduction state) and low (oxidation state) conductance levels equalizes over time. We observed a similar result when the molecule starts at the low conductance level (oxidation state, Supplementary Fig. 9).

The 2D $G$–$t$ histogram shown in Fig. 5c describes the evolution of the high and low conductance states over time, from which we extracted kinetic constants of redox switching at the single-molecule level. At a given time, we obtained a conductance histogram like the ones plotted in Fig. 5d, which shows the histogram at $t = 0.0$ and 0.1 s. The peak area reflects the probabilities of the molecule in reduction and oxidation states at the given time. Figure 5e plots the normalized peak area of the high conductance state (reduction state) versus time at

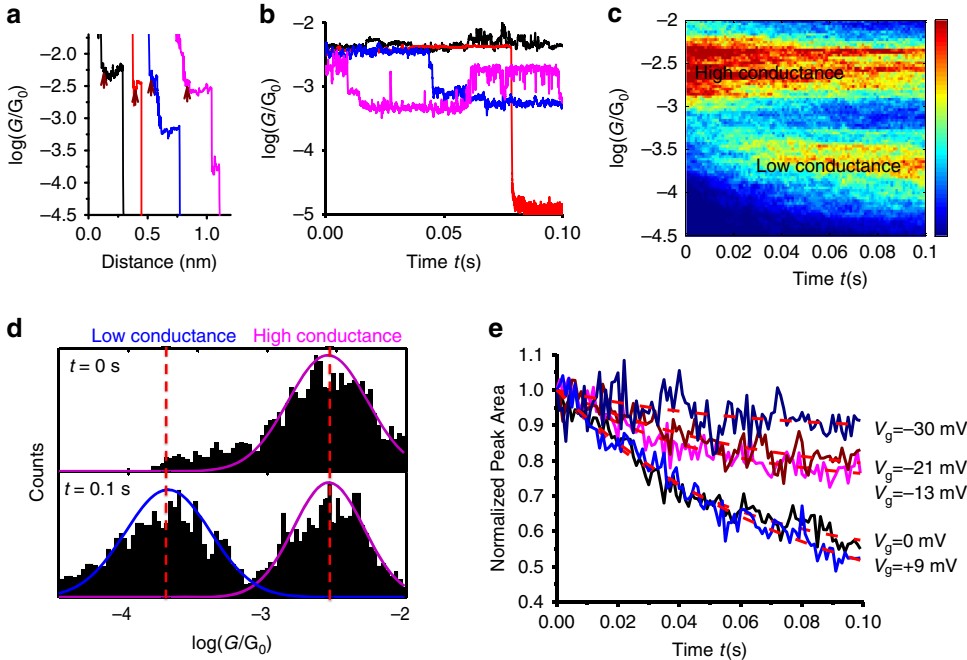

**Figure 5 | Kinetic analysis of redox reactions in single-DNA molecules.** (**a**) Conductance versus distance traces, each shows a plateau that corresponds to a DNA molecule bridged between the tip (source) and substrate (drain). The arrows mark the locations where the tip was fixed in position, and conducting switching events versus time were studied. Each trace leads to a G–t (conductance–time) trace with the same colour in **b**. (**b**) Three conductance switching behaviours: conductance stays at the high conductance level over the time window (black), conductance switches to the low conductance level and stays at that level (blue) and conductance switches back and forth between the two levels (magenta). Note that red trace shows that conductance drops to zero, due to the breakdown of the DNA junction. See text for more details. (**c**) 2D conductance (G) versus time (t) histogram with the gate voltage ($V_g$) set at $0.000 \pm 0.005$ V, showing two discrete conductance bands, and dependence on time. (**d**) Conductance histogram at $t = 0.0$ and $0.1$ s, showing transition from high conductance state to a mixture of high and low conductance states. (**e**) Normalized peak area of the high conductance peak versus time under different gate voltages, where the red dashed lines are the fitting of the curves with the rate equation (4).

different gate voltages. The peak area decays over time with a rate that depends on the gate voltage. The more positive is the gate voltage ($V_g$), the faster it decays, which is expected because the probability of Aq-DNA switching from the high conductance reduction state to the low conductance oxidation state increases with the gate voltage. This process can be described with a kinetic model[54] that is used in EC study of DNA charge transport[55,56],

$$\text{H}_2\text{Aq} \underset{k_b}{\overset{k_f}{\rightleftarrows}} \text{Aq} + 2e^- + 2\text{H}^+ \qquad (2)$$

where $k_f$ and $k_b$ are forward and backward rate constants, respectively, which are related by

$$k_f/k_b = K = \Gamma_{ox}/\Gamma_{red} \qquad (3)$$

where $K$ is the equilibrium constant that depends on $V_g$ according to equation (1). The probability of Aq-DNA remaining in the reduction state, $P_t(\text{red})$ can be expressed as

$$P_t(\text{red}) = \frac{k_f e^{-(k_b + k_f)t} + k_b}{k_b + k_f} \qquad (4)$$

By fitting the decay curve at $V_g = 0.000 \pm 0.005$ V (black curve in Fig. 5e) with equation (4), we found that $k_f = 9.8 \pm 0.3$ s$^{-1}$ and $k_b = 10 \pm 1$ s$^{-1}$. These values are within the range of charge transfer rate constants obtained from electrochemistry study on DNA with thiolate linkers. For example, the electron transfer rate constant of dsDNA (20 base pairs long) tethered with anthraquinone[36] was shown to be $1.3 \pm 0.3$ s$^{-1}$. Other electron transfer rate constants of dsDNA tethered with different redox active groups[37,57,58] ranges from 1 to 100 s$^{-1}$. The seemingly

surprising low electron transfer rate and high conductance value are probably due to the two different electron transfer channels in the present hoping dominated charge transport, as shown by Agostino Migliore and Abraham Nitzan's work[59]. One determines the redox state of Aq, which is the slow EC electron transfer rate, and the other channel dominates the conduction through the entire molecule. This finding is consistent with recent works by Zhou et al.[60] and by Venkatramani et al.[61], both concluded that relationship between the electron transfer rate and conductance depends on the transport mechanism. Using the rate constants, we obtained $\Gamma_{ox}/\Gamma_{red}$ with equation (3), and substituting it into equation (1) leads to $V_g = 0.000 \pm 0.002$ V, which is consistent with the actual applied gate voltage. We carried out the experiment at other gate voltages (Supplementary Fig. 10) and found the fitted and actual gate voltages agree with each other (Supplementary Table 2), which further confirms the kinetic model.

**Theoretical calculations of the energy diagrams**. The above analysis shows that the kinetic model developed for redox reactions can describe the observed conducting switching in Aq-DNA. However, the kinetic model cannot explain why the conductance in the reduction state is much greater than that in the oxidation state. To qualitatively understand the conductance difference between the two states, we carried out first principle calculations within density functional theory at the M06-2X/6-311 + G(p,d) level of theory[62,63]. This functional is suitable for our energy level and electronic coupling calculations, which involve non-covalent interactions, such as $\pi - \pi$ stacking.

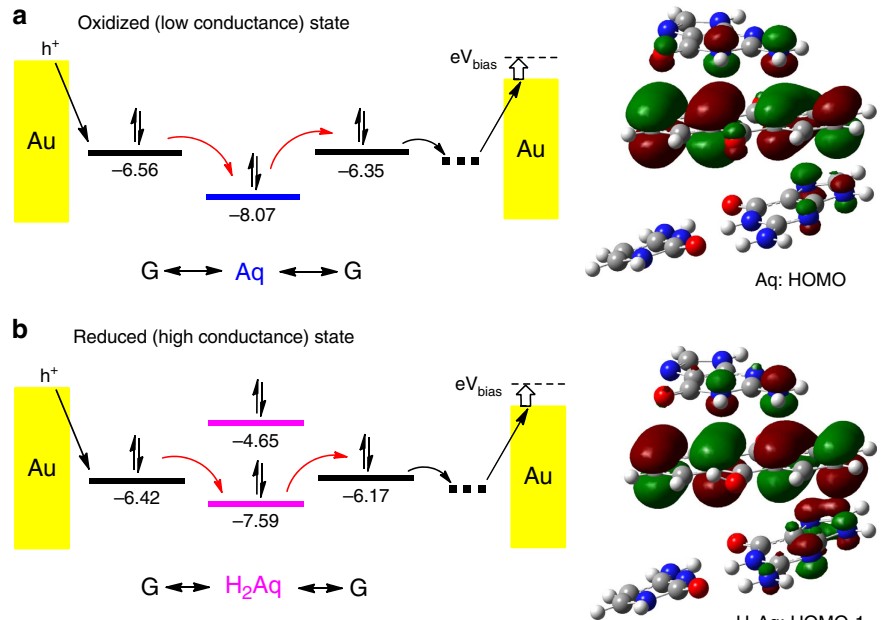

**Figure 6 | Energy diagram and molecular orbital spatial distribution. (a)** For oxidation state, HOMO level of Aq is the closest to the HOMO levels of Guanine. Hole hops from the left Guanine (non-paired) to Aq, then to the right Guanine (paired with C) as indicated by the red arrows. Molecular orbital spatial distribution indicates the HOMO level mainly localized on Aq. **(b)** For reduction state, HOMO-1 level of $H_2Aq$ is the closest to the HOMO levels of Guanine. Comparing with the oxidation state, the energy alignment between $H_2Aq$ and Guanines is better. Molecular orbital spatial distribution also indicates the HOMO-1 level mainly localized on $H_2Aq$. The unit is eV for all the energy levels.

We built two molecular fragments that represent the differences in our sequences, G-Aq-G:C (oxidation state) and G-$H_2Aq$-G:C (reduction state). Both systems were built based on the fragments of 2KK5 in Protein Data Bank (PDB)[17] where the Aq is located, with the DNA bases being replaced by our sequences using B-form DNA structure. Circular dichorism spectra show that this is the most probable DNA helical form in our experiments.

It is widely accepted that charge transport through DNA is due to hopping through the guanine HOMO[3,14,15], so we focused only on the Aq and $H_2Aq$ molecular levels that are the closest to the guanine HOMO level. Our quantum mechanical calculation shows that the Aq HOMO and the $H_2Aq$ HOMO-1 levels are the closest to the guanine HOMO level (Fig. 6). Furthermore, the molecular orbitals of the Aq HOMO and $H_2Aq$ HOMO-1 have similar spatial distributions; both are mainly localized at the anthraquinone moiety. Electronic couplings and energy levels were obtained based on the two-state model[15]. Our results show that Aq and $H_2Aq$ have similar coupling strengths with neighbouring guanine bases (0.52 and 0.05 eV for Aq; 0.36 and 0.12 eV for $H_2Aq$), but their energy level alignments are different. The $H_2Aq$ HOMO-1 level is ~0.34 eV closer to the guanine HOMO levels compared with the Aq HOMO level. We therefore suggest that the closer energy level alignment is the main reason for the higher conductance in the reduced state. In our previous works we have used ZINDO/S semiempirical method[15] to calculate energy levels and electronic couplings. We compared M06-2X/6-311 + G(p,d) and ZINDO/S (energy level ZINDO/S calculations, Supplementary Fig. 14) and found a good agreement between methods. Finally, we modified the position of the anthraquinone and calculated the energy levels and electronic couplings and found that our trend is robust suggesting that the energy alignment is indeed responsible for the difference in conductance between Aq and $H_2Aq$ (Supplementary Figs 15 and 16). Our observation is consistent with other reported results[24–26,64–66], which shows the conductance of anthraquinone in the reduction state is more conductive than that in the oxidation state.

**Discussion**

To verify that the Aq-DNA is bridged between the two electrodes via the two terminal thiolate groups as shown in Fig. 1a, we carried out the conductance measurement on Aq-DNA with either 3′-thiol or 5′-thiol modification (Supplementary Fig. 11). No peaks were revealed in the conductance histograms, indicating that both thiolate groups are necessary for forming molecular junctions and the Aq-DNA is linked to the tip and substrate electrodes via the Au-S bonds. The Aq moiety is located near one end of the molecule, and this end can either bind to the tip or the substrate electrodes (Fig. 3d). Since both the tip and the substrate are Au electrodes with a small potential difference (<0.1 V) controlled in the experiment, we expect similar results with either orientations. Furthermore, we measured current–voltage (I–V) curves and found that I–V curves and 2D conductance–voltage (G–V) histogram are symmetric (Supplementary Fig. 12), indicating the orientation of Aq-DNA in the molecule junction does not affect the measured conductance. This observation is consistent with the hopping model, which describes the resistance of the molecule as a sum of individual hopping steps[67,68].

Charge transport in redox molecules, including anthraquinone molecules with two thiolate groups[24], have been studied[20,25,33,34,44,45] under EC control, where the conductance was observed to change continuously with applied potential. This is in sharp contrast to the discrete two-level switching found in Aq-DNA. For a redox molecule directly connected to two electrodes, its conductance is directly related to the probability of oxidation (reduction) determined by the rates of electron transfer into and out of the molecule, which changes continuously with the potential[43,69]. In the present case, the

anthraquinone is either in the reduction or oxidation state as described by the Nernst equation, which determines the Aq-DNA to be either in the low or high conductance states, respectively. One possible explanation is that the oxidized or reduced anthraquinone provides a hopping site along the entire DNA sequence. To further explore the role of DNA in the charge transport, we carried out the gate-controlled conductance measurements on another anthraquinone-modified DNA sequence with longer length, Aq-DNA-2. Similar two-state conductance switching behaviour with smaller conductance values was observed (Supplementary Fig. 13), which further confirms the involvement of the DNA sequence in the charge transport of Aq-DNA. The experiment also demonstrates that one can tune the conductance of the switcher by changing the DNA sequence.

Our work demonstrates one can introduce an active control to DNA conductance by modifying a base with a redox group, and switch the DNA conductance reversibly between two levels by oxidizing or reducing the redox group with an EC gate. This strategy could be implemented in more sophisticated DNA nanostructures for active device building blocks. As the DNA conductance is an indicator of the molecule in the reduction or oxidation state, it is possible to study redox reaction kinetics at the single-molecule level by monitoring the DNA conductance.

## Methods

**DNA sample preparation.** Anthraquinone-modified oligonucleotide was purchased from Alpha DNA (high-performance liquid chromatograph (HPLC) purified). All the other oligonucleotides were purchased from Integrated DNA Technologies (HPLC purified). One of the oligonucleotides (Fig. 1b) was modified with 3′-thiol C3 S-S and 5′-thiol C6 S-S in its protected form. The oligonucleotides were dissolved in $18.4\,M\Omega \cdot cm$ deionized (DI) water to reach a concentration of $100\,\mu M$ and stored at $-20\,°C$. Sodium cacodylate trihydrate ($\geq 98\%$), magnesium perchlorate (ACS reagent, and $\geq 98\%$) and cacodylic acid ($\geq 98\%$) were purchased from SIGMA-Aldrich, and sodium perchlorate monohydrate (for HPLC, $\geq 99.0\%$) was purchased from Fluka. All the reagents were used without further purification. Multigene Mini Thermal Cycler (Model: TC-050-18) was used to anneal DNA solution samples. Cacodylate buffer (pH = 7.0) was prepared by dissolving 21.4 mg sodium cacodylate trihydrate, 22.3 mg magnesium perchlorate, 196.6 mg Sodium perchlorate monohydrate and 2 mg cacodylic acid in 10 ml $18.4\,M\Omega \cdot cm$ DI water. The oligonucleotide with thiolate linkers was deprotected with dithiothreitol (DTT) solution for 1 hour, then transferred to a spin column (Roche Applied Science quick spin column sephadex G-25) and centrifuged to remove DTT and the protection group. The oligonucleotide was then mixed with the complementary strand (Fig. 1b) with a stoichiometric ratio of 1:1 (calibrated by absorption intensity at 260 nm) and annealed by varying temperature from 80 to $8\,°C$ at the rate of $4\,min \cdot °C^{-1}$, and then kept at $4\,°C$ before measurements.

**Gel electrophoresis.** The electrophoretic measurement was performed at 200 V, and $22\,°C$ for 2.5 h using 50 pmol of each sample and with 8% nondenaturing polyacrylamide gel electrophoresis gels in $1 \times TAE$ (Tris base, acetic acid and Ethylenediaminetetraacetic acid) $Mg^{2+}$ buffer. The gels were subsequently stained with ethidium bromide and scanned in a Biorad Gel Doc XR+ system for sample visualization.

**Melting temperature and circular dichroism.** Melting temperature experiments were performed in a Varian Cary 300 Bio ultraviolet spectrophotometer with a Peltier thermal controller to determine melting temperature. 10 uM dsDNA were prepared with cacodylate buffer and annealed as for STM-BJ measurements, then heated at a rate of $0.2\,°C \cdot min^{-1}$ from 20 to $80\,°C$ with the absorbance at 260 nm recorded in 60s intervals. Melting temperature was obtained by fitting the melting temperature curves to a two-state thermodynamic model. Circular dichroism spectra were collected on a Jasco (Easton, MD) J-815 Spectropolarimeter from 320 to 220 nm with a scanning rate of $50\,nm\,min^{-1}$. The spectra were compiled by averaging the results from five scans, taken in cacodylate buffer solution at room temperature to replicate the environment during STM break junction experiments.

**STM break junction measurement.** Gold substrates were prepared by thermally evaporating $\sim 160\,nm$ of gold (99.999% purity, Alfa Aesar) onto freshly cleaved mica slides and annealed in ultra-high vacuum ($10^{-8}\,torr$) for 3 h. Before each experiment, the gold substrate was flame annealed for 1 min with a hydrogen flame.

The STM tip was freshly cut from gold wire (99.95% purity, Alfa Aesar) and coated with Apiezon wax to reduce the leakage current directly through aqueous solution[7]. All measurements were carried out in cacodylate buffer at room temperature ($22\,°C$). A small ( voltage was applied between the tip and substrate (5 mV, otherwise stated). As a control the STM break junction measurement was initially performed without DNA in cacodylate buffer, and the measured conductance histogram was found to be featureless (Supplementary Fig. 2a). Then 50 ul 5 uM double strand DNA was added to the buffer. A large number of current–distance traces ($\sim 4,000$) were recorded for each experiment, from which the conductance histogram was constructed with an algorithm described elsewhere[51]. To minimize noise, the algorithm counted only the traces showing counts exceeding a preset threshold in the histograms, and it selected 10–15% of the traces. For each double-strand DNA, the measurement was repeated three times to estimate the experimental error (see Supplementary Note 3 and Supplementary Table 3 for discussions). The EC gate-controlled measurements were performed under nitrogen atmosphere and the cacodylate buffer was purged by nitrogen (99.99% purity) for 30 min before use. The gate voltage was controlled by a biopotentiostat (Agilent). DNA was immobilized on the gold substrate by exposing the substrate in 10 uM DNA solution for 1 h, followed by rinsing with cacodylate buffer to remove non-bound DNA, and then filled with the buffer.

**Cyclic voltammetry.** Cyclic voltammetry was performed on the DNA modified gold substrate with a platinum wire as the counter electrode, a Ag/AgCl (in 1 M KCl solution) as reference electrode using an Autolab potentiostat. Ten repeated potential cycles of cyclic voltammograms were obtained for each sample with a typical sweeping rate of $100\,mV \cdot s^{-1}$ (or otherwise state. In addition to characterizing Aq-DNA immobilized on the gold substrate, cyclic voltammetry was performed before and after each STM break junction experiment to check the stability of the silver quasi-reference electrode, and the difference in the redox potential of Aq-DNA was taken as the error in the gate voltages (see Supplementary Fig. 3a for the discussion on the shifting of Ag quasi-reference electrode).

**Computational methods.** We performed quantum chemical calculations to obtain energy and electronic couplings of the hopping sites using molecular fragments, G-Aq-G:C and G-H$_2$Aq-G:C. Both systems were set up based on the 2KK5 of the PDB, which is conformed by a similar structure to ours with a terminal purine DNA base followed by an anthraquinone[17]. We obtained the Hamiltonian from a density functional theory calculations at the M06-2X/6-311+G(p,d) level of theory and divided it into their segments, which represent the hopping sites (G and Aq/H$_2$Aq). The energy and electronic coupling were calculated using the HOMO wave function of the Guanine and the neighbour occupied orbitals of the Aq/H$_2$Aq. These energy levels have been widely used as a reasonable approximation of the adiabatic wave function for the charge donor and acceptor[70].

**Data availability.** The data that support the findings of this study are available from the corresponding author upon reasonable request.

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

## Acknowledgements

We thank Shuoxing Jiang and Professor Hao Yan for the help with native polyacrylamide gel electrophoresis gel experiments. Financial support from the Office of Naval Research (N00014-11-1-0729) is gratefully acknowledged.

## Author contributions

L.X. performed the STM break junction measurements and electrochemical cyclic voltammetry study. J.L.P. performed the energy and electronic coupling calculations. M.A.R. and V.M. supervized the quantum mechanical calculations. N.T. conceived the experimental design and supervised the experiments. All authors contributed to the data analysis and writing.

## Additional information

**Competing financial interests:** The authors declare no competing financial interests.

**Publisher's note**: 

