## [Peer Review File · Nature Communications]

Reviewers' Comments:

Reviewer #1 (Remarks to the Author)

This manuscript by Xiang et al. represents an important advance in single molecule electronics and bio-electronics and will be of broad interest to a wide spectrum of Nature Communication readers. A significant conductance switching has been achieved in DNA through incorporation of anthraquinone replacing a DNA base and its electrochemical switching between redox states. Single molecule conductance is recorded as a function of electrochemical potential and it is shown that as the anthraquinone group is electrochemically reduced there is a large increase in conductance of the DNA strand. Interestingly, there is an abrupt change in conductance between the two redox states which is markedly different to other studies where electrochemical switching gives rise to a more gradual transition. A model for the switching between the two redox states is proposed to explain the reversible switching between two discrete states.

I have a number of points for the authors to address, which are mostly technical in nature:

(a) Page 3. The conductance values of both Aq-DNA and u-DNA appear very high. The conductance values are broadly comparable with the much shorter and fully conjugated 4,4' bipyridine. This is perhaps surprising looking at the molecular structures on page 23 (Figure 1) with both duplexes having a C3 combined with a C6 linker. The combined resistance of the linkers (C3 + C6, i.e. C9 in total) is much higher than the resistances of either Aq-DNA and u-DNA. Can this be explained within the theoretical modeling?

(b) On page 5 a value of 1.48 pmol/cm² is given for the surface coverage. What does this say about the orientation of DNA closely packed as vertically aligned or tilted cylinders?

(c) On page 7 stretching length is analyzed, which is explained to be "the average distance over which one can stretch a molecular junction before it becomes mechanically unstable". This is interesting but in addition, and perhaps more importantly, the complete junction breaking length (the length of the breaking of the molecular junction following the snap to contact of the gold electrodes) should also be presented. This would then show how much the molecule is pulled up in the junction following breaking of the metallic contact.

(d) On page 8 it is stated that "the hysteresis appears because the charge transfer rate between the electrode and Aq moiety is comparable to the sweeping rate". Could this be detailed through a quantitative comparison of these two values?

(e) The rate constant on page 11 is 10 s⁻¹. It would be good if this could be benchmarked against other measured values for electrochemically determined rate constants for anthraquinone systems (particularly surface attached).

(f) On page 11 it is stated that "this relatively low rate constant is limited by the thiolated linkers". It seems surprising that the rate is so low while the conductance is so high? Amatore and Mao in the publication "Do Molecular Conductances Correlate with Electrochemical Rate Constants? Experimental Insights" in JACS 2011 (J. Am. Chem. Soc. 2011, 133, 7509–7516) found that fast systems indeed give higher conductance. Also see the publications "The Single-Molecule Conductance and Electrochemical Electron-Transfer Rate Are Related by a Power Law" (DOI: 10.1021/nn401321k) and "Breaking the simple proportionality between molecular conductances and charge transfer rates" (DOI: 10.1039/c4fd00106k).

(g) On page 16 it is stated that "A large number of current–distance traces (~4,000) were recorded for each experiment, from which the conductance histogram was constructed with an algorithm described elsewhere". Please provide details of the percentage of traces selected by the algorithm.

(f) Figure 2d shows CVs for Aq-DNA. From these it would be possible to determine the electrochemical rate constant for the quasi-reversible process. This would be a very useful for independently determining the value for the electrochemical rate constant which can be feed back into the discussion.

(g) As a minor point the red histogram in Figure 1d is partially hidden by the blue one.

I am happy to recommend publication following these points being satisfactorily addressed.

Reviewer #2 (Remarks to the Author)

Review Tao Nat Comm August 2016 - AqDNA switch

A. Summary of the key results

The paper "Gate-controlled Conductance Switching in DNA" by Xiang et al. describes electrical transport measurements in Aq-DNA and unmodified DNA of two lengths. The transport is switched between two states using electrochemical gating. The measurements are done using STM break-junction and are controlled by many other experiments, well described in the MS and SI. The results are modeled and fitted theoretically.

The work is really beautiful, novel, well done and controlled and well written. I strongly and gladly recommend publication after minor revision, following my comments below..

Two general remarks: The conclusions by the model or other assumptions are stated too strongly. While the experimental results are clear and well established, the interpretation are a possibility. While I do not challenge the interpretations, I would therefore suggest to use a softer and milder wording in interpretations.

A second and technical point: The figures are not arranged and numbered in order of appearance, especially in the SI, which is sometimes confusing.

B. Originality and interest: if not novel, please give references

The paper is original and very interesting to the molecular electronics community.

C. Data & methodology: validity of approach, quality of data, quality of presentation

The data is nice, complete and is enough to draw the conclusions, though in a milder way. The results are compared and controlled by a variety of methods and are generally well presented (see my comment above). Furthermore, both the groups of Tao and the theory groups of Ratner and Mujica are very well experienced in exactly this type of measurements and calculations/modeling, further strengthening the validity of the results.

D. Appropriate use of statistics and treatment of uncertainties

Both the statistics and errors are well treated (See sections in SI).

E. Conclusions: robustness, validity, reliability

The conclusions and interpretations are reasonable and well explained in the text. The results are well controlled and therefore robust and reliable.

F. Suggested improvements: experiments, data for possible revision

I find no need for significant improvements or needed experiments. Minor comments below.

G. References: appropriate credit to previous work?

In some cases the authors refer to hopping, as well established for charge transfer experiments between donor and acceptor in solution experiments. While these models explain well the hopping for those experiments, they can not be a direct reference in experiments in which the molecule is attached to two metal electrodes where the Fermi levels (chemical potential) is fixed, the potential landscape over the molecule is modified (even for low voltage as in this experiment) and many charges pass through the molecules. In these cases the ref should be to hopping in similar systems. This was demonstrated by few experiments, including by the Tao group. So referencing of this point should be corrected.

Otherwise the paper is well referenced.

H. Clarity and context: lucidity of abstract/summary, appropriateness of abstract, introduction and conclusions

The abstract and intro are clear. I would allow to extend the abstract and add a sentence on the theory and model already there. Intro and summary are fine.

Additional minor comments:

A. The choice of the sequence should be explained, e.g., around line 51 or elsewhere.

B. Line 52 - "and other experiments": Either remove or outline.

C. Line 62 - "Thousands of current traces": add ~4000. It is mentioned later in the methods but I think it should appear here too for the reader to have a magnitude of the repeats. Up to you.

D. Line 82: The authors claim that it is a B-form DNA. Although the experiments are done in buffer (BTW - not like that of X-ray), I doubt that the actual measured molecules, which are stretched,

indeed retain the B-form structure. The CDs seem with difference. I am not sure that these CDs are enough to determine that these short DNA molecules have a B-form. This is especially in doubt during the experiment, in which the molecules are stretched and unlikely to retain the B-form in this relevant situation.

E. Line 201: The number of digits in the number and in error are not the same....

F. Line 225: same for times.

G. Line 249: same.

H. Line 263-4: Same as my comment D above.

I. Line 265: refs relate to charge transfer and should refer to charge transport (see my comment above).

J. Line 289: how is the molecule orientation determined? Why the I-Vs are not asymmetric (the 3' and 5' have C3 and C6 linkers....

Reviewer #3 (Remarks to the Author)

Nature Communications manuscript NCOMMS-16-16576-T The manuscript NCOMMS-16-16576-T, by Limin Xiang et al., entitled "Gate-controlled Conductance Switching in DNA", reports evidence that the conductance of a DNA-based molecular device can be switched in a controlled manner by chemical treatment with a redox species. This is a breakthrough in the quest for DNA-based molecular nanotechnology. The authors first demonstrate (Fig. 1) their STM breakjunction technique to measure DNA conductance between two electrodes. Fig. 1d also illustrates that the measurement approach is able to resolve between different molecular species. Then, for the hybrid species Aq:DNA they show redox activity by cyclic voltammetry, which is a standard convincing approach (Fig. 2). In Fig. 2 the authors also show the higher stability of Aq:DNA than that of bare DNA (μ -DNA) in terms of higher melting temperature, as well as the similar helix conformation by circular dichroism. The core results are presented in Fig. 3: STM breakjunction measurements performed in the presence of a third electrode that acts as gate electrode show that the conductance can be switched by tuning the gate voltage relative to the redox potential of the anthraquinone species; with a series of control experiments and deep analysis of the data, the authors convincingly show that the conductance switch is associated to the Aq redox activity. By the Nernst electrochemistry model, it is shown that the redox process is a two-electron process, as it is known for Aq. The time-dependent analysis of the conductance illustrates different behaviors (Fig. 5), compatible with the redox states populations. Kinetic modeling at different gate voltages further reinforces the interpretation in terms of Aq redox activity and yields values for charge transfer rate constants: such values are consistent with published reports on DNA systems. Last, by electronic structure calculations, the authors show that two different conductance values in the reduced and oxidized Aq states are due to the different alignments of the Aq electronic energy levels to the HOMO of guanine. The presentation is clear and fluent, leading the reader to more and more convincing evidence. The work is original and of broad interest. The data and methodology are accurately described and discussed: all the methods are valid to attain the goal, though I have a concern on the electronic structure approach, which I point out later. The statistical analysis is satisfactory. The conclusions are soundly based on the data and data analysis. References are appropriately cited. Based on the above assessment, I recommend publication of the manuscript in Nature Communications, after the authors consider my specific recommendations listed below. Most of them are really minor and optional, but I invite the authors to seriously consider my recommendations on the electronic structure calculations.

- Abstract and Introduction. Both in the first sentence of the abstract and at the bottom of page 1 in the introduction, the authors write that

much is known about the charge transport properties of DNA, while conductance switch is needed to turn it into an electronic component. It seems they mean that things would be ready for technology transfer if conductance switch is demonstrated, while this is not the case. Even if conductance switch is demonstrated, there is still much work to do before being able to exploit charge transport in DNA to build nanodevices. I suggest to add a short comment on this, or to change the current formulation, to tone down the claim. • Page 2 line 53. The end of the sentence should probably be “no other structures are present under the experimental conditions” (it seems to me that the verb is missing). • Page 3 line 67. I think it would be more fair to write both conductance values in the same order of magnitude, 10^{-4} . What is the given error? Is it the width of the statistical distribution of each peak? • Page 5 lines 113 and 115. I would also like to see these conductance values expressed in the same order of magnitude. And why is the error not reported here? Aren't the values obtained in the same manner as from Fig. 1d? • Page 5 line 123. “all the molecule become”, the verb should be without final s. • Page 5 line 126. I think measurements should be plural. • Page 8 lines 181-182. First I read “E is the gate voltage” and then “gate voltage = $E - E_{ox/red}$ ”, which seems inconsistent. Maybe it should be written that E is the absolute gate voltage, while the relevant one is the gate voltage relative to the redox potential. • Page 8 lines 183 and 185. Faraday should be with capital F in line 183. In line 185, “of for” is redundant. • Page 12 lines 261-277. This appears to me as a “weak” part in the manuscript, as it were done in a hurry after all the experimental part was ready. I overall accept that, but it can be improved. It is written: “We built two molecular fragments, [...], based on the canonical B-DNA and the structure 2KK5 [...]”. The statement is somewhat confusing, because it seems that the oxidation state was constructed from canonical B-DNA and the reduction state from 2KK5, but I don't think this was the case. I guess the information from B-DNA and 2KK5 was used together to construct both fragments. But 2KKA contains a helical shape, so what additional structural information is needed from canonical B-DNA? Why is the short fragment with an unpaired G, an Aq and a GC pair is sufficient to represent the experimental system? Is this a single electronic structure calculation for a single structure of each redox state? If so, was the structure optimized? Or was it representative of dynamics? ZINDO/S is a semiempirical approximation of the electronic structure, while more precise methods are affordable for DNA nowadays. Especially if structural optimization was performed, is it reliable at this level of theory? Is the order of energy levels reliable? This is crucial for the presented interpretation. In the section on Computational Methods, Ref. 68 is cited to justify the electronic structure treatment: that references addresses a search over many structures, and I accept that to screen multiple conformations a simplified electronic method is a “forced” choice. However, for two structures only something more accurate can be done. At least I would like to see a comparison to hybrid-DFT for the same structures, and a comparison of the energy level alignment for different structures. One more concern: at line 271 electronic couplings are reported, but it is not clear from the text if these values are computed within this work or are taken from elsewhere. I guess from other text parts that they have been computed, but this can be stated more clearly. My recommendation would be to add a discussion in the Supporting Information to motivate the choice of the electronic structure level of theory, ideally showing comparison to a more precise method and for at least two different structures for each redox state, to validate the conclusions on the energy level alignment. This is easily doable in less than a week on nowadays computers, and would certainly reinforce the conclusions. I like the work presented in this manuscript very much, so I think it is worth performing a little refinement to make it “perfect”! • Pages 23-24 figure 1 and caption. For panel b, the caption should start with “From left to right”, or indicate clearly the left, middle and right parts. In the

center, Aq is red, while in the sides it is blue. I recommend the use of the same color for Aq everywhere. If the authors choose to have it blue as in the left scheme, then it should be changed in the central structure, which is easily doable using any visualization software for PDB files. Otherwise, the colors in the side schemes should be changed, and also in Fig. 1d and in Supporting Figures 3, 4, 7, 8, 9. • Pages 26 figure 3. The red fitting lines are poorly visible, they should at least be thicker, and maybe a clearer color can be considered. • Pages 28 figure 5. The grey dots in panel a are not distinguishable, maybe enlargement solves the problem. In the caption of panel a, one would expect a comment of what are the four different colors. It becomes clear in the discussion of panel b, but it could be anticipated.

Reviewers' comments:

Reviewer #1 (Remarks to the Author):

This manuscript by Xiang et al. represents an important advance in single molecule electronics and bio-electronics and will be of broad interest to a wide spectrum of Nature Communication readers. A significant conductance switching has been achieved in DNA through incorporation of anthraquinone replacing a DNA base and its electrochemical switching between redox states. Single molecule conductance is recorded as a function of electrochemical potential and it is shown that as the anthraquinone group is electrochemically reduced there is a large increase in conductance of the DNA strand. Interestingly, there is an abrupt change in conductance between the two redox states which is markedly different to other studies where electrochemical switching gives rise to a more gradual transition. A model for the switching between the two redox states is proposed to explain the reversible switching between two discrete states.

I have a number of points for the authors to address, which are mostly technical in nature:

Response: We thank reviewer for the positive comments.

(a) Page 3. The conductance values of both Aq-DNA and u-DNA appear very high. The conductance values are broadly comparable with the much shorter and fully conjugated 4,4' bipyridine. This is perhaps surprising looking at the molecular structures on page 23 (Figure 1) with both duplexes having a C3 combined with a C6 linker. The combined resistance of the linkers (C3 + C6, i.e. C9 in total) is much higher than the resistances of either Aq-DNA and u-DNA. Can this be explained within the theoretical modeling?

Response: Thanks for the interesting note. Unlike 4,4' bipyridine, charge transport in DNA is primarily due to hopping (rather than tunneling), which decreases much slower with distance than that in a tunneling process, leading to long range charge transport in DNA. Similar conductance values (10^{-4} to $10^{-3} G_0$) were reported for dsDNA with C3 or C6 thiolate groups (Arte's, J. M. et al. *Nat. Commun.* **6, 8870 (2016); Hihath, J. et al. *Proc. Natl. Acad. Sci. U. S. A.* **102**, 16979 (2015); Xu, B. et al. *Nano Lett.* **4**, 1105 (2004)).**

(b) On page 5 a value of 1.48 pmol/cm² is given for the surface coverage. What does this say about the orientation of DNA closely packed as vertically aligned or tilted cylinders?

Response: The surface coverage of 1.48 pmol/cm², corresponds to 112 nm² surface area per DNA molecule, which indicates that the DNA molecules is likely tilted.

(c) On page 7 stretching length is analyzed, which is explained to be "the average distance over which one can stretch a molecular junction before it becomes mechanically unstable". This is interesting but in addition, and perhaps more importantly, the complete junction breaking length (the length of the breaking of the molecular junction following the snap to contact of the gold electrodes) should also be presented. This would then show how much the molecule is pulled up in the junction following breaking of the metallic contact.

Response: We have updated the stretching length in Supplementary figure 7, which is the length measured starting from the abrupt current drop before the plateau (breakdown of Au-Au

contact) to that after plateau (breakdown of DNA junction). The stretching length reflects how long one can stretch the DNA, and the average value agrees with a previous study (Bruot C. et al. *ACS Nano* **9**, 88 (2015)).

(d) On page 8 it is stated that “the hysteresis appears because the charge transfer rate between the electrode and Aq moiety is comparable to the sweeping rate”. Could this be detailed through a quantitative comparison of these two values?

Response: We have expanded the discussion in the revised manuscript.

(e) The rate constant on page 11 is 10 s^{-1} . It would be good if this could be benchmarked against other measured values for electrochemically determined rate constants for anthraquinone systems (particularly surface attached).

Response: We agree, and compared it with the rate constants for several other systems in the revised manuscript.

(f) On page 11 it is stated that “this relatively low rate constant is limited by the thiolated linkers”. It seems surprising that the rate is so low while the conductance is so high? Amatore and Mao in the publication “Do Molecular Conductances Correlate with Electrochemical Rate Constants? Experimental Insights” in *JACS* 2011 (*J. Am. Chem. Soc.* 2011, 133, 7509–7516) found that fast systems indeed give higher conductance. Also see the publications “The Single-Molecule Conductance and Electrochemical Electron-Transfer Rate Are Related by a Power Law” (DOI: 10.1021/nn401321k) and “Breaking the simple proportionality between molecular conductances and charge transfer rates” (DOI: 10.1039/c4fd00106k).

Response: This is an excellent point, and we added a discussion in the revised manuscript. The system discussed by Amatore and Mao is in the regime of tunneling. In the present hopping dominated charge transport, there are two different electron transfer channels as shown by Agostino Migliore and Abraham Nitzan’s work (*J. Am. Chem. Soc.* 135, 9420–9432 (2013)). One determines the redox state of Aq, which is the slow electrochemical electron transfer rate, and the other channel dominates the conduction through the entire molecule. This finding is consistent with recent works by Wierzbinski et al. (*J. Am. Chem. Soc.* **133**, 7509–7516 (2011)) and by Venkatramani et al. (*Faraday Discuss.*, **174**, 57-78 (2014)), both concluded that relationship between the electron transfer rate and conductance depends on the transport mechanism. In fact, Wierzbinski et al. showed that PNA duplexes with the lowest charge transfer rates have the highest conductance.

(g) On page 16 it is stated that “A large number of current–distance traces ($\sim 4,000$) were recorded for each experiment, from which the conductance histogram was constructed with an algorithm described elsewhere”. Please provide details of the percentage of traces selected by the algorithm.

Response: We have included this information to the manuscript.

(f) Figure 2d shows CVs for Aq-DNA. From these it would be possible to determine the electrochemical rate constant for the quasi-reversible process. This would be a very useful for

independently determining the value for the electrochemical rate constant which can be feed back into the discussion.

Response: Accurate electrochemical rate constant determination by the Laviron method requires performing CV over a wide sweep rate range (including ultra-fast sweeping). We found that fast CV sweeping induce instability in the present Aq-DNA system, and our CV measurement was limited to slow sweep rates (Supplementary figure 3). Although we were not able to accurately measure the rate constant from the CVs, our rate constant of 10 s^{-1} is within the reasonable range of other reported values.

(g) As a minor point the red histogram in Figure 1d is partially hidden by the blue one. I am happy to recommend publication following these points being satisfactorily addressed.

Response: We thank the reviewer for catching this flaw, and we have fixed it in figure 1d.

Reviewer #2 (Remarks to the Author):

Review Tao Nat Comm August 2016 - AqDNA switch

A. Summary of the key results

The paper "Gate-controlled Conductance Switching in DNA" by Xiang et al. describes electrical transport measurements in Aq-DNA and unmodified DNA of two lengths. The transport is switched between two states using electrochemical gating. The measurements are done using STM break-junction and are controlled by many other experiments, well described in the MS and SI. The results are modeled and fitted theoretically.

The work is really beautiful, novel, well done and controlled and well written. I strongly and gladly recommend publication after minor revision, following my comments below..

Response: We thank the reviewer for the positive feedback.

Two general remarks: The conclusions by the model or other assumptions are stated too strongly. While the experimental results are clear and well established, the interpretation are a possibility. While I do not challenge the interpretations, I would therefore suggest to use a softer and milder wording in interpretations.

Response: We have re-worded the interpretation of the experimental results.

A second and technical point: The figures are not arranged and numbered in order of appearance, especially in the SI, which is sometimes confusing.

Response: We have re-arranged the figures according to the order presented in the main text.

B. Originality and interest: if not novel, please give references

The paper is original and very interesting to the molecular electronics community.

Response: We thank the reviewer for speaking highly of the originality and interest of our work.

C. Data & methodology: validity of approach, quality of data, quality of presentation

The data is nice, complete and is enough to draw the conclusions, though in a milder way. The

results are compared and controlled by a variety of methods and are generally well presented (see my comment above). Furthermore, both the groups of Tao and the theory groups of Ratner and Mujica are very well experienced in exactly this type of measurements and calculations/modeling, further strengthening the validity of the results.

Response: We appreciated the comments.

D. Appropriate use of statistics and treatment of uncertainties
Both the statistics and errors are well treated (See sections in SI).

Response: We appreciated the comments.

E. Conclusions: robustness, validity, reliability
The conclusions and interpretations are reasonable and well explained in the text. The results are well controlled and therefore robust and reliable.

Response: We appreciated the comments.

F. Suggested improvements: experiments, data for possible revision
I find no need for significant improvements or needed experiments. Minor comments below.

Response: We appreciated the comments.

G. References: appropriate credit to previous work?
In some cases the authors refer to hopping, as well established for charge transfer experiments between donor and acceptor in solution experiments. While these models explain well the hopping for those experiments, they can not be a direct reference in experiments in which the molecule is attached to two metal electrodes where the Fermi levels (chemical potential) is fixed, the potential landscape over the molecule is modified (even for low voltage as in this experiment) and many charges pass through the molecules. In these cases the ref should be to hopping in similar systems. This was demonstrated by few experiments, including by the Tao group. So referencing of this point should be corrected. Otherwise the paper is well referenced.

Response: We have fixed the references accordingly.

H. Clarity and context: lucidity of abstract/summary, appropriateness of abstract, introduction and conclusions
The abstract and intro are clear. I would allow to extend the abstract and add a sentence on the theory and model already there. Intro and summary are fine.

Response: We have emphasized the theory and model in the abstract.

Additional minor comments:

A. The choice of the sequence should be explained, e.g., around line 51 or elsewhere.

Response: We have added the DNA sequence information to the main text.

B. Line 52 - "and other experiments": Either remove or outline.

Response: We have fixed the typo.

C. Line 62 - "Thousands of current traces": add ~4000. It is mentioned later in the methods but I think it should appear here too for the reader to have a magnitude of the repeats. Up to you. "Thousands of current traces"

Response: We appreciated the comment and added the information to the text.

D. Line 82: The authors claim that it is a B-form DNA. Although the experiments are done in buffer (BTW - not like that of X-ray), I doubt that the actual measured molecules, which are stretched, indeed retain the B-form structure. The CDs seem with difference. I am not sure that these CDs are enough to determine that these short DNA molecules have a B-form. This is especially in doubt during the experiment, in which the molecules are stretched and unlikely to retain the B-form in this relevant situation.

Response: We agree with the reviewer, and have rephrased the statement about B-form conformation for DNA molecules during STM break junction experiments.

E. Line 201: The number of digits in the number and in error are not the same....

Response: We have fixed this issue.

F. Line 225: same for times.

Response: We have fixed this issue too.

G. Line 249: same.

Response: We have fixed this issue.

H. Line 263-4: Same as my comment D above.

Response: We agree with the reviewer and clarified this point.

I. Line 265: refs relate to charge transfer and should refer to charge transport (see my comment above).

Response: We have fixed the references.

J. Line 289: how is the molecule orientation determined? Why the I-Vs are not asymmetric (the 3' and 5' have C3 and C6 linkers....

Response: The orientation of the molecule is un-controlled in the system. We do not expect asymmetric I-Vs because for hopping transport the total resistance is a sum of resistance from the individual hopping sites, which is independent on the orientation. Furthermore, asymmetry

would not affect conductance values under low bias voltages (Supplementary Figure 12).

Reviewer #3 (Remarks to the Author):

Report attached

Nature Communications manuscript NCOMMS-16-16576-T

The manuscript NCOMMS-16-16576-T, by Limin Xiang et al., entitled “Gate-controlled Conductance Switching in DNA”, reports evidence that the conductance of a DNA-based molecular device can be switched in a controlled manner by chemical treatment with a redox species. This is a breakthrough in the quest for DNA-based molecular nanotechnology. The authors first demonstrate (Fig. 1) their STM breakjunction technique to measure DNA conductance between two electrodes. Fig. 1d also illustrates that the measurement approach is able to resolve between different molecular species. Then, for the hybrid species Aq:DNA they show redox activity by cyclic voltammetry, which is a standard convincing approach (Fig. 2). In Fig. 2 the authors also show the higher stability of Aq:DNA than that of bare DNA (u-DNA) in terms of higher melting temperature, as well as the similar helix conformation by circular dichroism. The core results are presented in Fig. 3: STM breakjunction measurements performed in the presence of a third electrode that acts as gate electrode show that the conductance can be switched by tuning the gate voltage relative to the redox potential of the anthraquinone species; with a series of control experiments and deep analysis of the data, the authors convincingly show that the conductance switch is associated to the Aq redox activity. By the Nernst electrochemistry model, it is shown that the redox process is a two-electron process, as it is known for Aq. The time-dependent analysis of the conductance illustrates different behaviors (Fig. 5), compatible with the redox states populations. Kinetic modeling at different gate voltages further reinforces the interpretation in terms of Aq redox activity and yields values for charge transfer rate constants: such values are consistent with published reports on DNA systems. Last, by electronic structure calculations, the authors show that two different conductance values in the reduced and oxidized Aq states are due to the different alignments of the Aq electronic energy levels to the HOMO of guanine. The presentation is clear and fluent, leading the reader to more and more convincing evidence. The work is original and of broad interest. The data and methodology are accurately described and discussed: all the methods are valid to attain the goal, though I have a concern on the electronic structure approach, which I point out later. The statistical analysis is satisfactory. The conclusions are soundly based on the data and data analysis. References are appropriately cited. Based on the above assessment, I recommend publication of the manuscript in Nature Communications, after the authors consider my specific recommendations listed below. Most of them are really minor and optional, but I invite the authors to seriously consider my recommendations on the electronic structure calculations.

Response: We appreciate the positive remarks.

- Abstract and Introduction.

Both in the first sentence of the abstract and at the bottom of page 1 in the introduction, the authors write that much is known about the charge transport properties of DNA, while conductance switch is needed to turn it into an electronic component. It seems they mean that things would be ready for technology transfer if conductance switch is demonstrated, while this is not the case. Even if conductance switch is demonstrated, there is still much work to do

before being able to exploit charge transport in DNA to build nanodevices. I suggest to add a short comment on this, or to change the current formulation, to tone down the claim.

Response: We appreciate the suggestion and rephrased the sentence.

- Page 2 line 53.

The end of the sentence should probably be “no other structures are present under the experimental conditions” (it seems to me that the verb is missing).

Response: We have fixed the typo.

- Page 3 line 67.

I think it would be more fair to write both conductance values in the same order of magnitude, 10^{-4} . What is the given error? Is it the width of the statistical distribution of each peak?

Response: We have changed the notation of the conductance values for easier comparison. The error was estimated by repeating the conductance measurement at least three times (the standard deviation of the repeated measurements). We have included a discussion to the supplementary discussion 3 and added supplementary table 3.

- Page 5 lines 113 and 115.

I would also like to see these conductance values expressed in the same order of magnitude. And why is the error not reported here? Aren't the values obtained in the same manner as from Fig. 1d?

Response: We agree, and have made the suggested change to include errors. The values were obtained in a similar manner as that in the previous comment.

- Page 5 line 123.

“all the molecule become”, the verb should be without final s.

Response: We have fixed this typo.

- Page 5 line 126.

I think measurements should be plural.

Response: We have fixed this typo.

- Page 8 lines 181-182.

First I read “E is the gate voltage” and then “gate voltage = $E - E_{ox/red}$ ”, which seems inconsistent. Maybe it should be written that E is the absolute gate voltage, while the relevant one is the gate voltage relative to the redox potential.

Response: We have fixed this error.

- Page 8 lines 183 and 185.

Faraday should be with capital F in line 183. In line 185, “of for” is redundant.

Response: We have fixed this typo.

- Page 12 lines 261-277.

This appears to me as a “weak” part in the manuscript, as it were done in a hurry after all the experimental part was ready. I overall accept that, but it can be improved.

It is written: “We built two molecular fragments, [...], based on the canonical B-DNA and the structure 2KK5 [...]”. The statement is somewhat confusing, because it seems that the oxidation state was constructed from canonical B-DNA and the reduction state from 2KK5, but I don’t think this was the case. I guess the information from B-DNA and 2KK5 was used together to construct both fragments. But 2KKA contains a helical shape, so what additional structural information is needed from canonical B-DNA? Why is the short fragment with an unpaired G, an Aq and a GC pair is sufficient to represent the experimental system? Is this a single electronic structure calculation for a single structure of each redox state? If so, was the structure optimized? Or was it representative of dynamics? ZINDO/S is a semiempirical approximation of the electronic structure, while more precise methods are affordable for DNA nowadays. Especially if structural optimization was performed, is it reliable at this level of theory? Is the order of energy levels reliable? This is crucial for the presented interpretation. In the section on Computational Methods, Ref. 68 is cited to justify the electronic structure treatment: that references addresses a search over many structures, and I accept that to screen multiple conformations a simplified electronic method is a “forced” choice. However, for two structures only something more accurate can be done. At least I would like to see a comparison to hybrid-DFT for the same structures, and a comparison of the energy level alignment for different structures. One more concern: at line 271 electronic couplings are reported, but it is not clear from the text if these values are computed within this work or are taken from elsewhere. I guess from other text parts that they have been computed, but this can be stated more clearly. My recommendation would be to add a discussion in the Supporting Information to motivate the choice of the electronic structure level of theory, ideally showing comparison to a more precise method and for at least two different structures for each redox state, to validate the conclusions on the energy level alignment. This is easily doable in less that a week on nowadays computers, and would certainly reinforce the conclusions. I like the work presented in this manuscript very much, so I think it is worth performing a little refinement to make it “perfect”!

Response: We appreciate the reviewer comments, and have modified the text to clarify the calculations. Both the oxidized and reduced forms of the molecule were built based on the 2KK5 structure of the PDB with DNA molecules in B-form. However, one base (adenine) in the 2KK5 structure was replaced by a guanine to simulate the structure studied here.

According to the reviewer’s suggestions, we have also performed another higher level of calculation based on DFT at the M06-2X/6-311+G(p,d) level, and the results agree with the original ZINDO/S calculations. We have included the results in the main text, and moved the ZINDO/S calculations to the supporting information (Supplementary Fig. 14).

Finally, we have included additional calculations of different structures for the oxidation and reduced states to test the robustness of our calculations and possible dynamic fluctuations and their effects on energy alignment and coupling strength (Supplementary Fig. 15-16). We have calculated the electronic coupling obtained from the Hamiltonian matrix in the DFT calculation using a partition method with the two-level model. Our results show that the difference in the

energy alignment and coupling strength between the two states depends weakly on the position of Aq moiety, which further supports our conclusions. We have updated these results in the main text.

- Pages 23-24 figure 1 and caption.

For panel b, the caption should start with “From left to right”, or indicate clearly the left, middle and right parts. In the center, Aq is red, while in the sides it is blue. I recommend the use of the same color for Aq everywhere. If the authors choose to have it blue as in the left scheme, then it should be changed in the central structure, which is easily doable using any visualization software for PDB files. Otherwise, the colors in the side schemes should be changed, and also in Fig. 1d and in Supporting Figures 3, 4, 7, 8, 9.

Response: We have fixed the figure caption and the color of Aq.

- Pages 26 figure 3.

The red fitting lines are poorly visible, they should at least be thicker, and maybe a clearer color can be considered.

Response: We have fixed Figure 3 to make the fitting more clearly visible.

- Pages 28 figure 5.

The grey dots in panel a are not distinguishable, maybe enlargement solves the problem. In the caption of panel a, one would expect a comment of what are the four different colors. It becomes clear in the discussion of panel b, but it could be anticipated.

Response: We have fixed these plots.

Reviewers' comments:

Reviewer #1 (Remarks to the Author):

I have gone carefully through the responses to the Referees. In the main I find the replies reasonable but a few important points need firming up, or some extra text needs to be added to the manuscript in the replies to Reviewer #1. Otherwise, I find this an interesting and important manuscript which is well suited to publication in Nature Communications.

(1) For Point (a), since the conductance values and mechanism of charge transfer are central to the whole study, I believe that the response here should be included in a suitable place in the main manuscript, pointing out the unexpectedly high conductance values (when benchmarked against bipyridine for instance) and the implications deduced by the authors that hopping must be operating.

(2) For Point (b), the value of 112 nm² per DNA molecule seems a low coverage unless the tilting is very large? I suggest including this in the manuscript together with comments on the packing, orientation and tilting implications.

(3) For Point (c), it is still not completely clear to me how long the molecular junction is when it breaks? In supplementary figure 7a a distance scale is given from 1.2 to 1.4 nm, what is this distance? I would suggest more text is needed to explain this point clearly.

(4) Point (f), the statement "In fact, Wierzbinski et al. showed that PNA duplexes with the lowest charge transfer rates have the highest conductance", needs clarification. In this article (Faraday Discuss., 2014, 174, 57–78) Figure 1, the opposite is shown for both ds and ss PNAs, i.e. as the rate constant increases, the conductance increases.

(5) Related to point (4) above, the new references on page 12 (58 and 59) have been confused.

(6) Point (f). Just in noting, there are methods other than Laviron by which rate constant can be estimated through CV fitting.

Reviewer #2 (Remarks to the Author):

I read the paper and browsed the SI. It looks fine to me after the revision. Can be accepted now.

Reviewer #3 (Remarks to the Author):

The authors have satisfactorily addressed my comments and those of the other reviewers. I recommend the revised manuscript for publication in Nature Comm

Reviewers' comments:

Reviewer #1 (Remarks to the Author):

I have gone carefully through the responses to the Referees. In the main I find the replies reasonable but a few important points need firming up, or some extra text needs to be added to the manuscript in the replies to Reviewer #1. Otherwise, I find this an interesting and important manuscript which is well suited to publication in Nature Communications.

(1) For Point (a), since the conductance values and mechanism of charge transfer are central to the whole study, I believe that the response here should be included in a suitable place in the main manuscript, pointing out the unexpectedly high conductance values (when benchmarked against bipyridine for instance) and the implications deduced by the authors that hopping must be operating.

Response: We agree with the reviewer, and have added the discussions on Page 3 in the revised manuscript.

(2) For Point (b), the value of 112 nm² per DNA molecule seems a low coverage unless the tilting is very large? I suggest including this in the manuscript together with comments on the packing, orientation and tilting implications.

Response: Yes, the surface coverage of DNA is low compared to a compact layer. We have added this information and discussion into the revised manuscript (page 5).

(3) For Point (c), it is still not completely clear to me how long the molecular junction is when it breaks? In supplementary figure 7a a distance scale is given from 1.2 to 1.4 nm, what is this distance? I would suggest more text is needed to explain this point clearly.

Response: The distance scale in supplementary figure 7a is the position of the STM tip measured from the piezoelectric scanner, and only the change in the value is meaningful. The length of the plateau (indicated by the distance between the two black dash lines in supplementary figure 7a) is how long a molecular junction can be stretched before it breaks down. We have clarified this in the main text (page 7) and supplementary figure 7a caption.

(4) Point (f), the statement "In fact, Wierzbinski et al. showed that PNA duplexes with the lowest charge transfer rates have the highest conductance", needs clarification. In this article (Faraday Discuss., 2014, 174, 57–78) Figure 1, the opposite is shown for both ds and ss PNAs, i.e. as the rate constant increases, the conductance increases.

Response: We thank the reviewer for correcting the error, and have removed the error in the revised manuscript (page 12).

(5) Related to point (4) above, the new references on page 12 (58 and 59) have been confused.

Response: We have fixed the references on Page 12.

(6) Point (f). Just in noting, there are methods other than Laviron by which rate constant can be estimated through CV fitting.

Response: We thank the reviewer for this helpful information.